# The Liver in Heart Failure: From Biomarkers to Clinical Risk

**DOI:** 10.3390/ijms242115665

**Published:** 2023-10-27

**Authors:** Nadia Aspromonte, Isabella Fumarulo, Lucrezia Petrucci, Bianca Biferali, Antonio Liguori, Antonio Gasbarrini, Massimo Massetti, Luca Miele

**Affiliations:** 1Department of Cardiovascular and Thoracic Sciences, Catholic University of the Sacred Heart, 00168 Rome, Italy; isabella.fumarulo01@icatt.it (I.F.); massimo.massetti@policlinicogemelli.it (M.M.); 2Department of Cardiovascular and Thoracic Sciences, A. Gemelli University Policlinic Foundation IRCCS, 00168 Rome, Italy; 3Department of Translational Medicine and Surgery, Catholic University of the Sacred Heart, 00168 Rome, Italy; lucreziapetruccimed@gmail.com (L.P.); bbiferali@gmail.com (B.B.); antonio.liguori@guest.policlinicogemelli.it (A.L.); antonio.gasbarrini@policlinicogemelli.it (A.G.); luca.miele@policlinicogemelli.it (L.M.); 4Department of Medical and Surgical Sciences, A. Gemelli University Policlinic Foundation IRCCS, 00168 Rome, Italy

**Keywords:** biomarkers, prognostic scores, heart failure, liver stiffness measurement, elastography, liver fibrosis

## Abstract

Heart failure (HF) is a clinical syndrome due to heart dysfunction, but in which other organs are also involved, resulting in a complex multisystemic disease, burdened with high mortality and morbidity. This article focuses on the mutual relationship between the heart and liver in HF patients. Any cause of right heart failure can cause hepatic congestion, with important prognostic significance. We have analyzed the pathophysiology underlying this double interaction. Moreover, we have explored several biomarkers and non-invasive tests (i.e., liver stiffness measurement, LSM) potentially able to provide important support in the management of this complex disease. Cardiac biomarkers have been studied extensively in cardiology as a non-invasive diagnostic and monitoring tool for HF. However, their usefulness in assessing liver congestion in HF patients is still being researched. On the other hand, several prognostic scores based on liver biomarkers in patients with HF have been proposed in recent years, recognizing the important burden that liver involvement has in HF. We also discuss the usefulness of a liver stiffness measurement (LSM), which has been recently proposed as a reliable and non-invasive method for assessing liver congestion in HF patients, with therapeutic and prognostic intentions. Lastly, the relationship between LSM and biomarkers of liver congestion is not clearly defined; more research is necessary to establish the clinical value of biomarkers in assessing liver congestion in HF patients and their relationship with LSM.

## 1. Introduction

Heart failure (HF) is a clinical syndrome that occurs when heart dysfunction (structural or functional) leads to elevated intracardiac pressures and/or inadequate cardiac output at rest and/or during exercise, resulting in symptoms and signs. Right heart failure (RHF) occurs when there is dysfunction of the right heart structures, including the right ventricle, tricuspid valve apparatus, right atrium, and pericardium, leading to reduced perfusion of the lungs at normal central venous pressures [1,2]. RHF can occur acutely due to conditions such as pulmonary embolism, acute respiratory distress syndrome, right ventricular myocardial infarction, or cardiac tamponade, or it can occur chronically due to conditions such as pulmonary hypertension, congenital heart disease, cardiomyopathy, right heart valve disease, or constrictive pericarditis. Acquired forms of left heart failure, both with preserved ejection fraction (HFpEF) and reduced ejection fraction (HFrEF), are the most common causes of RHF [3,4].

## 2. Liver in Heart Failure

The liver is vulnerable to circulatory disorders due to its complex vascular anatomy and high metabolic activity. The degree and characteristics of liver injury depend on the blood vessels involved and the degree to which the injury is related to passive congestion or impaired perfusion [5]. Any cause of RHF can cause hepatic congestion, including constrictive pericarditis, mitral stenosis, tricuspid regurgitation, and cardiomyopathy (Figure 1). Tricuspid regurgitation, in particular, may be associated with severe hepatic congestion due to the transmission of right ventricular pressure directly into the hepatic veins. Congestive hepatopathy (CH) occurs when chronic passive venous congestion results from elevated central venous pressure (CVP) in RHF, which is transmitted to the hepatic (central) veins [6,7]. This results in pre-sinusoidal dilation, decreased hepatic artery blood flow, and decreased arterial oxygen saturation, which can ultimately lead to irreversible congestive liver fibrosis and cardiac cirrhosis (Figure 2). CH is characterized by dilation of the lobular hepatic veins and hepatic sinusoids, perisinusoidal edema, acinar steatosis, and heterogeneous fibrosis [8,9].

The burden of CH has not been well established yet, particularly in a general population setting where there is a shortage of epidemiologic studies [10].

Only the most severe cases typically manifest hepatic complications, such as jaundice, which may be mistaken for biliary obstruction [11]. In acute heart failure, jaundice and a significant increase in serum aminotransferases may simulate acute viral hepatitis [12,13]. Patients may also experience right upper quadrant discomfort due to straining of the liver capsule and ascites [14,15]. On clinical examination, the liver border is typically firm, smooth, and somewhat tender, and ascites may occur. Hepatojugular reflux is usually present and can help differentiate liver congestion from primary intrahepatic liver disease or Budd–Chiari syndrome. Imaging studies such as ultrasound and cross-sectional CT can reveal hepatomegaly, hepatic vein dilatation with diminished respiratory variation and retrograde portal vein flow with phasic changes, venous dilation, reflux of contrast into the inferior vena cava (IVC), hepatic veins in the arterial phase, and delayed parenchymal enhancement in the venous phase [16,17,18]. Hepatic congestion due to elevated CVP progressively leads to liver fibrosis up to cirrhosis, which characterizes the end-stage of CH [19].

Hepatic fibrosis is the most relevant prognostic factor in individuals affected by a chronic liver disease (CLD), exerting a profound influence on long-term outcome and mortality [20,21]. Liver biopsy is still considered the gold standard for detecting liver injury and fibrosis stage. However, it has several limitations: invasiveness, risk of complications (bleeding or infections), sample variability, and interobserver variability [22]. Due to these limitations, in the last decade, research efforts have focused extensively on the development of alternative non-invasive tests (NITs). The ideal diagnostic approach should have qualities of precision (accurate measurement), consistency (yielding consistent results upon repeated evaluations), and adaptability (responsive to changes in fibrosis levels over time). NITs developed for liver fibrosis assessment could be divided into blood-based biomarkers (ELF, fibrosis-4 score FIB4, NAFLD Fibrosis Score NFS) and elastography techniques [23]. Elastography techniques aim to obtain a liver stiffness measurement (LSM) as a surrogate marker of liver fibrosis. It is still an indirect assessment of liver fibrosis, in fact liver stiffness can be influenced by other concurrent pathophysiological processes, including inflammation, passive venous congestion, portal hypertension, and biliary obstruction [19,24,25]. There are four main technologies developed to perform liver elastography: vibration-controlled transient elastography (VCTE, FibroScan), acoustic radiation force impulse (ARFI), shear wave elastography (SWE, either point SWE or bidimensional 2D-SWE), and magnetic resonance elastography (MRE) [26]. The results of transient elastography are expressed in kPa and can vary from 2.5 to 75 kPa. The cut-offs for the diagnosis of significant fibrosis or cirrhosis depend on the underlying liver disease and the elastography technique used, but in the clinical setting, LSM values (VCTE) from >7 kPa for significant fibrosis and >11 to 14 kPa for cirrhosis are commonly used [23].

In this scenario, accurate assessment of fibrosis progression is essential for classifying patient with CLD, including CH, and therefore guiding treatment decisions. A liver biopsy can help in the diagnosis of congestive liver disease, especially when the diagnosis is uncertain or when assessing the severity of histological damage [27]. However, it is rarely performed due to its invasiveness, associated complications, and limited sampling. Instead, NITs validated for chronic liver diseases, such as the FIB-4 and fibrotest/fibroSURE, have been attempted to characterize fibrosis in congestive hepatopathy (CH) [28,29]. However, data in patients with heart disease have shown poor correlation between blood-based NITs, LSM, and actual liver fibrosis. In fact, LSM is heavily influenced by venous congestion, and in CH, it could not be considered a reliable marker of liver fibrosis [10].

Nevertheless, congestive heart disease, and consequently congestive hepatopathy, are chronic conditions that often occur in patients at an advanced age. It is not uncommon for CLD, irrespectively of the etiology, to have a silent clinical course until the development of cirrhosis and its complications such as ascites and portal hypertension. Therefore, it is appropriate in the clinical assessment of patients with suspected CH to investigate the presence of comorbidities that may have caused or contributed to CLD before or along with heart disease [30]. It appears mandatory in such patients to exclude the presence of chronic viral infections (HBV; HCV), to assess the extent of alcohol consumption, to exclude storage diseases (hemochromatosis) or autoimmune liver diseases, and to assess the presence of metabolic comorbidities underlying steatohepatitis (diabetes, hypertension, and obesity) [31].

## 3. Liver Biomarkers as Diagnostic and Prognostic Tools in Heart Diseases

Cardiac biomarkers have been studied extensively in cardiology as a non-invasive diagnostic and monitoring tool in patients with HF [32,33]. However, their usefulness in assessing liver congestion in HF patients is still under debate [14,15].

On the other side, the use of NITs has massively increased in hepatology to estimate the severity of hepatic fibrosis in almost all etiologies of liver disease (Figure 3) [28,29]. 

Liver involvement seems to have an important role in HF patients, especially in the final stages, potentially accelerating the negative progression of the disease [7,10]. In recent years, several prognostic scores based on liver biomarkers in patients with HF have been proposed (Table 1). 

The composite model for end-stage liver disease, excluding INR (MELD-XI), is a robust scoring system of liver function obtained from total bilirubin and creatinine, associated with poor prognosis in HF patients [37,39]. 

The common use of anticoagulant treatment in patients with HF led to the exclusion of the INR value from the validated score. MELD-XI score shows some controversial results in estimating hepatic fibrosis in patients with CH, especially in patients after the Fontan procedure. The correlation between MELD-XI and liver fibrosis in post-Fontan patients has been demonstrated in a retrospective study conducted on 70 patients, while it has not been confirmed in other studies with smaller sample size [40,41]. 

The NAFLD fibrosis score (NFS) and the FIB-4 scores, obtained from age, body mass index (BMI), and laboratory parameters such as platelets count, albumin, and transaminases, may predict the new onset of atrial fibrillation (AF) in patients with HFpEF [42].

The FIB-4 index turned out to be also a significant predictor for cardiovascular events in HFpEF [34]. 

The fibrosis-5 (FIB-5) index, which includes albumin, alkaline phosphatase, aspartate transaminase, alanine aminotransferase, and platelet count, showed further better prognostic values than FIB-4 in patients hospitalized with HF [35]. 

In the DAPA-HF trial, bilirubin concentration was an independent predictor of worse outcomes: participants in the highest bilirubin tertile had more severe HFrEF, a greater burden of AF, but less diabetes [38]. In an acute HF (AHF) setting, type III procollagen peptide levels may be able to predict adverse outcomes as a biomarker of liver dysfunction [36].

LSM has been proposed as a non-invasive method for assessing liver fibrosis in HF patients [43], but the accuracy of the method may be invalidated in CH where blood liver congestion results in at least modestly elevated LSM. Therefore, it becomes difficult to distinguish congestion from underlying fibrosis in CH. 

Some studies have shown a correlation between LSM and biomarkers such as serum bilirubin, gamma-glutamyl transferase (GGT), alkaline phosphatase (ALP), and the ratio of serum albumin/globulin in HF patients [44]. Nevertheless, the clinical usefulness of these biomarkers for assessing liver congestion in HF patients is still unclear and requires further investigation. Additionally, biomarkers such as N-terminal pro-B-type natriuretic peptide (NT-proBNP) and high-sensitivity troponin T (hsTnT) have been researched for HF diagnosis and monitoring, but their correlation with LSM and liver congestion in HF patients still needs to be determined. Overall, more research is necessary to establish the clinical value of biomarkers in assessing liver congestion in HF patients and their relationship with LSM.

## 4. Correlation between Non-Invasive Measurement of CVP and LSM

Nowadays, liver stiffness is known to reflect CVP [45]. Increased right atrial pressure causes liver congestion in most patients with heart failure syndrome due to the non-elastic capsule where the liver is wrapped. Liver congestion leads to an increase in liver stiffness [45,46]. 

The measurement of liver stiffness is performed with non-invasive methods such as VCTE, ARFI, and SWE. Therefore, standardizing this correlation will provide a non-invasive way to assess CVP. This is the reason why this topic has gained increasing attention (Table 2). 

In 2014, Taniguchi et al. first investigated LSM assessed by elastography as a non-invasive surrogate for a right atrial pressure (RAP) measurement in patients with HF [46]. Thirty-one patients with HF and without structural liver disease were enrolled; LSM with FibroScan, actual RAP with right-sided cardiac catheterization, and estimated RAP with echocardiography were measured. This study showed that the area under the curve of LSM for the identification of RAP > 10 mm Hg was 0.958 (95% CI 0.757 to 0.994, *p* < 0.0001), which was significantly greater than that of the two-dimensional echocardiographic IVC parameters (area under the curve = 0.800, 95% CI 0.604 to 0.913, *p* < 0.0001). Based on the ROC curve analysis, the optimal cut-off value for LSM for the detection of RAP > 10 mm Hg was 10.6 kPa.

LSM, which closely depends on the right-heart filling pressure, has been shown to be helpful in the evaluation of right ventricular (RV) function in the pre- and post-left ventricular assist device (LVAD) periods. Kashiyama et al. and Nishi et al. both analyzed this correlation by assessing LSM with FibroScan in 30 and 55 patients, respectively, before they underwent LVAD implantation [47,48]. LSM assessment was performed pre- and post-operatively, and the results were analyzed in correlation with the perioperative status. LSM successfully decreased after LVAD implantation, reflecting the effects of left ventricular (LV) unloading and the consequent decrease in RV afterload.

In line with the previously mentioned studies, Potthoff et al. evaluated the importance of LSM measured by ARFI in patients with HF who underwent LVAD implantation [49]. 28 patients with HF were enrolled. Before LVAD implantation, all patients underwent LSM by ARFI. According to the initial LSM values, patients were divided into two groups and followed up after 21 days (T1) and after 485 ± 136 days (T2). Analysis showed a significant decrease in the LSM values at T1 (*p* < 0.001) and T2 (*p* < 0.001), with respect to baseline.

In addition, in 2016, Yoshitani et al. investigated the relations between CVP and liver and kidney stiffness [45]. This study included controls (10) and HF patients (38). Liver and kidney stiffness were measured by ARFI (virtual touch quantification method) after and before treatment. LSM was significantly higher in the HF group than in the control group (1.17 ± 0.13 vs. 2.03 ± 0.91 m/s, *p* < 0.001), but there were no significant differences in kidney stiffness between the groups (2.14 ± 0.30 vs. 2.20 ± 0.60 m/s, *p* = 0.686). Therefore, CVP was defined as an independent predictive factor for increased liver stiffness in HF patients, and multivariate analysis showed a linear trend correlation (R = 0.636, *p* = 0.014). Important results also came from the LSM assessment after the treatments, where liver stiffness showed a significant decrease.

Studies on the pediatric population are also available. Terashi et al. and Jalal et al. both evaluated whether liver stiffness could be considered a non-invasive and reliable method for the assessment of CVP in children with heart diseases. Terashi et al. enrolled 79 patients (age < 20) and measured liver stiffness with SWE of the liver [50]. On the other hand, Jalal et al. enrolled 60 patients (median age 7.4 ± 5.5 years old) and measured liver stiffness with VCTE (FibroScan) [51]. Both studies confirmed what had already been shown in studies on adults. Moreover, they showed a significant correlation between LSM and CVP in children with congenital heart disease (R = 0.776, *p* < 0.001 vs. R = 0.68, *p* < 0.001, respectively).

After analyzing these studies, it is clear that liver stiffness measurement (LSM) could play a significant role in managing patients with heart failure, both in adults and children. While LSM is widely used in hepatology to assess liver fibrosis, its significance should be expanded to other fields, including cardiology, where it could serve as a reliable and non-invasive marker of hepatic congestion or right-sided filling pressure in HF patients. However, further studies are necessary to establish standardized protocols for using LSM to correlate with CVP and its implementation in clinical practice.

## 5. Elastography Can Demonstrate Decongestion in Patients with Heart Failure

Several studies have shown that measuring B-type natriuretic peptide (BNP) levels can be used as a prognostic factor in HF and to assess the benefit of HF treatment (Table 3). However, after an acute exacerbation, BNP normalization does not always follow a predictable pattern. Therefore, several studies have used LSM to define decongestion in patients with HF.

Many studies have compared LSM at hospitalization and discharge in patients admitted for HF, including those under medical therapy and those treated with device implants or cardiac surgery (LVAD, atrial septal defect closure). Patients with heart failure with reduced and preserved ejection fraction were evaluated using vibration-controlled transient elastography and point shear wave elastography. Almost all studies have shown a significant reduction in LSM after diuretic therapy, especially the study conducted by Milloning et al., which demonstrated excellent outcomes in patients subjected to medical therapy (median LSM at admission 40.7 (6.1–51.3) kPa and LSM 17.8 (3.3–33.2 kPa) before discharge) [52].

Furthermore, in these studies, median LSM was correlated with NTproBNP, right atrial pressure (RAP), and right ventricular pressure on the echocardiogram. In fact, Yoshitani et al. found that LSM and NTproBNP were both significantly reduced after sufficient diuresis [45]. Compared to other non-invasive markers of congestive HF, LSM more accurately demonstrates decongestion. Yoshitani et al. showed that body weight, LSM, and BNP all significantly decreased after diuretic therapy compared to total bilirubin, AST, ALT, and GGT.

Colli et al. demonstrated that liver stiffness values, like NTproBNP levels, tend to decrease in patients with acute decompensated HF after treatment, along with clinical improvement [53]. Alegre et al. had similar results: after clinical compensation, liver stiffness decreased in all patients in group acute HF [55]. 

Hopper et al. demonstrated that increased LSM correlated with increased bilirubin, GGT, and alkaline phosphatase in left-sided HF, right-sided HF, and acute decompensated HF groups but did not show a significant change in LSM after adequate diuresis (median LSM 11.2 kPa before diuresis treatment and 9.5 kPa after diuresis treatment) [54]. Sakamoto et al. demonstrated that shear wave velocity decreased after treatment (from 2.01 ± 0.61 to 1.62 ± 0.49 m/s; *p* = 0.026), while the liver fibrosis index did not change (from 1.21 ± 0.29 to 1.26 ± 0.27; *p* = 0.664) [57].

In patients with atrial septal defect (ASD), increased volume and pressure in the right atrium and right ventricle have been shown to increase LSM. For example, in a study conducted by Küçükosmanoğlu et al., LSM values assessed by point SWE were significantly increased in ASD patients with closure indication and Eisenmenger syndrome compared to patients without ASD closure indication [59]. Pekoz et al. demonstrated that LSM can be used as an objective follow-up parameter in addition to classic echocardiography in patients treated with ASD occluder devices. Among the 66 patients included (38 female, 28 male), in patients who underwent ASD closure, after a 1-year follow-up, LSM and liver size were significantly decreased [58].

Almost all of the cited studies agree on the utility of LSM as a superior tool for defining decongestion in patients with HF, considering that liver markers vary and are typically unreliable, despite larger shifts in body volume. However, there is an important limitation: the inability to determine accurate reference ranges to assess for adequate decongestion. While most studies show a significant decrease in LSM after diuresis, a standard has not been established, and it is unclear whether residual abnormalities in LSM reflect congestion or underlying fibrosis or whether there may be discordance between liver stiffness and congestion in HF. Future studies should establish an LSM cut-off to demonstrate effective decongestion in patients with HF.

## 6. Correlation between LSM and Adverse Outcomes in HF Patients

In recent years, various groups have investigated the prognostic value of LSM in HF patients, providing interesting results in terms of the correlation between LSM and adverse outcomes (Table 4).

Saito et al. performed LSM in 105 patients with acute decompensated heart failure (ADHF) using FibroScan on admission [60]. After a median follow-up of 153 days, those with higher LSM (≥8.8 kPa) had a higher incidence of primary endpoints of cardiovascular death and readmission for HF, with LSM being the only independent risk factor for cardiac events.

Omote et al. analyzed 70 ADHF patients, this time using point SWE/ARFI to obtain LSM on admission [61]. Patients with higher LSM (>1.50 m/s) had a poor prognosis after a median follow-up of 272 days. In addition to LSM, systolic blood pressure has also been identified as an independent risk factor for composite endpoints of all-cause death or worsening HF.

Taniguchi et al. performed LSM using FibroScan in 171 HF patients before discharge, with a subsequent median follow-up of 203 days [62]. High LSM was an independent risk factor for worse outcomes, since patients with LSM > 6.9 kPa (corresponding to an estimated right atrial pressure of 7.1 mm Hg) had a significantly higher incidence of cardiac death or readmission for HF. Furthermore, on ROC analysis, LSM > 10.1 kPa was identified as the optimal cut-off for predicting the occurrence of short-term cardiac events (sensitivity of 0.73; specificity of 0.90).

Soloveva et al. performed LSM in 149 ADHF patients using FibroScan both on admission and before discharge [56]. In both cases, higher LSM (>13 kPa on admission and >5 kPa at discharge) was related to a higher incidence of adverse events, but only discharge LSM has been identified as an independent risk factor for composite endpoints (all-cause death, heart transplant, or readmission for HF) at a median follow-up of 289 days.

Wang et al. evaluated LSM by FibroScan in 53 HF patients at discharge, using LSM > 6.9 kPa as the cut-off for the higher LSM group [63]. Once again, LSM was an independent risk factor for death or readmission for HF after a median follow-up of 730 days; in addition, tricuspid annular plane systolic excursion (TAPSE) was also an independent risk factor.

Panchani et al. performed LSM by 2D-SWE in 49 ADHF patients on admission, using a very different cut-off for the higher LSM group, up to >39.8 kPa [65]. After a median follow-up of 365 days, the higher LSM group had worse outcomes, with LSM resulting in being an independent risk factor for composite endpoints of death, LVAD, heart transplant, and rehospitalization. In detail, each 1 kPa increase in LSM was associated with a 1% increase in the incidence rate of readmission.

De Ávila et al. analyzed 85 out-patient HF patients, performing LSM using FibroScan [66]. On the ROC analysis, a cut-off point of 5.9 kPa had the best accuracy to predict the primary outcome of cardiovascular death or HF hospitalization, with a sensitivity of 80% and specificity of 64.1%, after a median follow-up of 219 days. Recently, two groups have focused on decompensated HFpEF patients only, also in this specific subpopulation, and their results are in agreement with the studies above. Saito et al. performed LSM using ultrasound 2D-SWE on discharge in 80 acute decompensated HFpEF patients [64]. Patients with higher LSM (>10.2 kPa) had worse outcomes after a median follow-up of 212 days, with LSM identified as an independent risk factor for composite endpoints (all-cause death; readmission for HF). Zhang et al. evaluated LSM using FibroScan in 150 decompensated HFpEF patients on admission and then followed them up for a mean of 197 days [43]. The group with LSM > 8.30 kPa had a poor prognosis. In addition to LSM, atrial fibrillation, New York Heart Association (NYHA) class, and NT-proBNP were independent risk factors for major adverse cardiac events (cardiovascular death, malignant arrhythmia, acute myocardial infarction, stroke, and rehospitalization for HF).

All the cited studies agree on the utility of LSM as a predictive prognostic factor in patients with HF across the entire spectrum of the disease, from HFrEF to HFpEF. At the moment, however, there is still no standardization in the method, as each group obtained LSM with different techniques and used different cut-offs to stratify risk classes. Furthermore, many of these studies evaluated patients with ADHF without having a previous baseline LSM value to compare with. Taking into account interindividual variability, it would be useful to perform a measurement of LSM at least once in HF patients, simultaneously with the measurement of RAP in basal conditions. This would provide a reference value that could be used during any subsequent hospitalizations, comparing it with the new LSM measurement, in order to guide diuretic therapy. Moreover, it would be important to define the timing of LSM. In some studies, it was obtained on admission, while in others, it was obtained on discharge. A single study (Soloveva et al. [56]) investigated both options, observing that only discharge LSM was an independent risk factor for composite endpoints. Accordingly, therapeutic response with adequate decongestion during hospitalization may be an index of a better prognosis, but this would require a reference value of LSM at baseline, as mentioned above. Finally, we must unfortunately remember that not all patients are eligible for these techniques; their applicability (especially VCTE) may be hindered by obesity, ascites, and other pathological liver conditions.

In conclusion, in the future, LSM could really become an aid in clinical practice for better prognostic stratification of HF patients, but these emerging techniques need more extensive validation, and an accurate standardization is needed.

## 7. Conclusions

Heart failure is a multisystemic disease which involves multiple organs, with the liver particularly vulnerable to congestion. In recent years, several liver biomarkers and derived prognostic scores in patients with HF have been proposed. At the same time, LSM has been proposed as a non-invasive method for assessing liver congestion in HF patients, with therapeutic and prognostic intentions. The relationship between LSM and biomarkers of liver congestion is not clearly defined; more research is necessary to establish the clinical value of biomarkers in assessing liver congestion in HF patients and their relationship with LSM. 

## Figures and Tables

**Figure 1 ijms-24-15665-f001:**
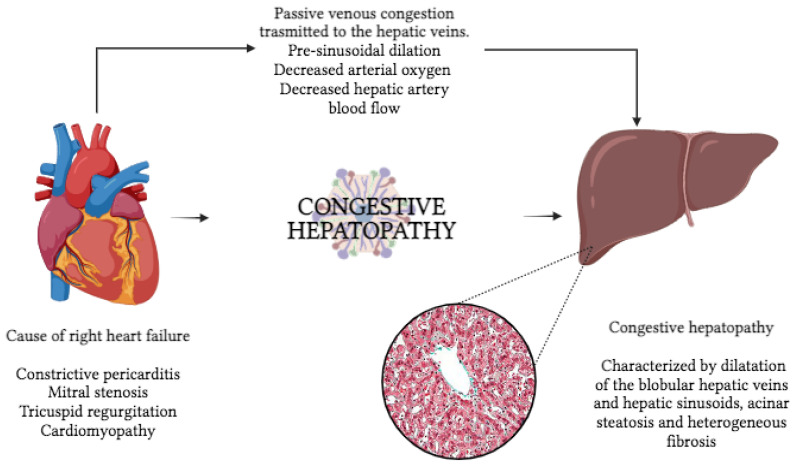
Pathophysiology of hepatic congestion in Heart Failure. RHF can have different etiologies (i.e., constrictive pericarditis, mitral stenosis, tricuspid regurgitation, or cardiomyopathy); whatever the cause, RHF leads to passive venous congestion transmitted to hepatic veins, ultimately causing congestive hepatopathy. RHF: right heart failure.

**Figure 2 ijms-24-15665-f002:**
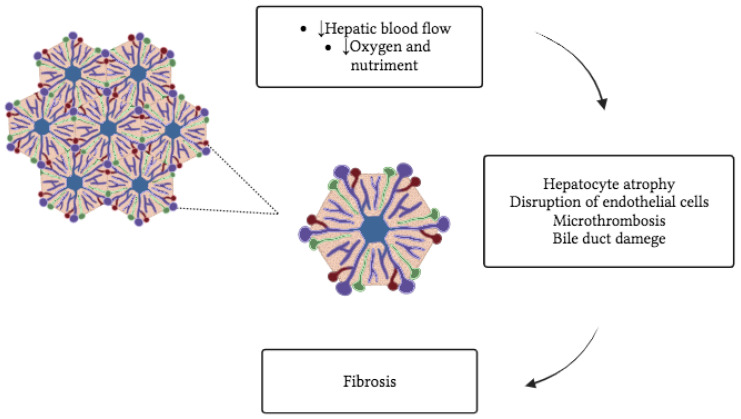
Physiopathological changes in congestive hepatopathy. CH is characterized by pre-sinusoidal dilation, decreased (↓) hepatic artery blood flow and decreased (↓) arterial oxygen saturation, which can ultimately lead to irreversible congestive liver fibrosis and cardiac cirrhosis. Key to the colors used: green for bile ducts, red for hepatic arterial vessels, purple for portal vessels. CH: Congestive hepatopathy.

**Figure 3 ijms-24-15665-f003:**
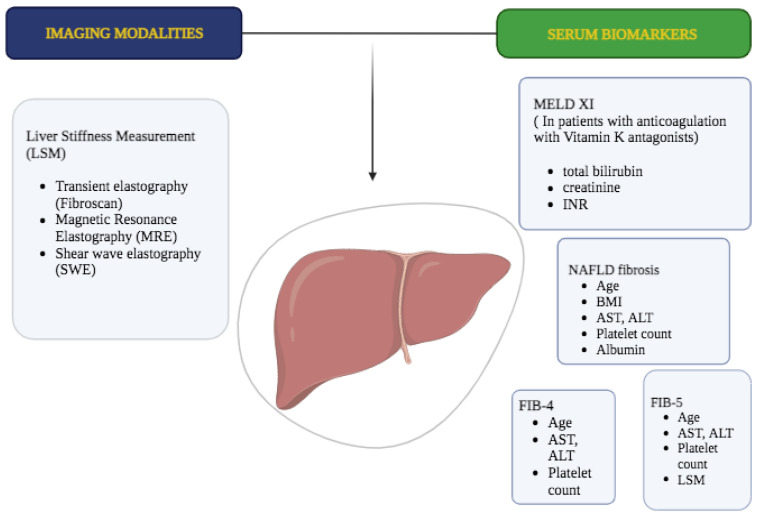
Hepatic fibrosis: Non-invasive techniques based on imaging modalities and serum biomarkers.

**Table 1 ijms-24-15665-t001:** Prognostic scores based on liver biomarkers in patients with HF.

Studies	Biomarker	Study Population	N	Cut-Off for Biomarker Group	Composite Endpoints	N° of Events	Mean FU Time (Days)	Independent Risk Factor (s) for Event
Takae et al. [34]	FIB-4	HF	704	>2.67 (on admission)	Total CV events			
		HFrEF	83			19	695	
		HFmrEF	117			26	632	
		HFpEF	504			237	1159	FIB-4
Maeda et al. [35]	FIB-5	HF	906	<−8.20 (on discharge)	Cardiac death; readmission for HF	320	152	FIB-5
Shirakabe et al. [36]	P3P	AHF admitted in ICU	643	>1.2 U/mL (on admission) >16.09%	All-cause death; readmission for HF	229	365	P3P

	PVS					307		PVS
Abe et al. [37]	MELD-XI	Decompensated HF	562	>10	Cardiac death; all-cause death	62	471	age
								reduced HF
						42		MELD-XI
Adamson et al. [38]	Bilirubin	HFrEF	4720	>1.0 mg/dL	CV death, worsening of HF, or all-cause death	885	720	bilirubin
	ALP		4729	>120 IU/L				
	ALT		4714	>35 IU/L				
	AST		4681	>35 IU/L				
Yang et al. [39]	MELD-XI	VAD	255	>17	CV death	48	365	MELD-XI

**Table 2 ijms-24-15665-t002:** Correlation between non-invasive measurement of CVP and LSM.

Studies	Modality	Study Population	N	LSM	Measured CVP	Correlation (r)	*p* Value
Taniguchi et al. [46]	FibroScan	Decompensated HF	31	8.5 (5.3–12.0) kPa	9.0 (5.0–12.0) mm Hg	0.95	<0.001
Nishi et al. [47]	FibroScan	LVAD recipients	30	13.3 ± 13 kPa	8.8 ± 6.9 mm Hg	0.515	<0.01
Kashiyama et al. [48]	FibroScan	LVAD recipients	55	12.7 ± 13.1 kPa	7.4 ± 5.0 mm Hg	0.52	<0.01
Potthoff et al. [49]	ARFI	LVAD recipients	28	2.50 ± 0.92 m/s	14.0 ± 6.0 mm Hg	0.793	0.001
Yoshitani et al. [45]	ARFI	Decompensated HF	38	2.03 ± 0.91 m/s	11.8 ± 5.4 mm Hg	0.636	0.014
Terashi et al. [50]	Ultrasound SWE	Children with congenital heart diseases	79	/	5.7 ± 3.5 mm Hg	0.776	<0.001
Jalal et al. [51]	FibroScan	Children/adults with congenital heart diseases	96 (60/36)	5 (2.8–47.2) m/s [4.6 (3–21)/6.1 (2.8–47.2)] m/s	6 (3–20) mm Hg [6 (3–15)/7 (3/20)] mm Hg	0.75 (0.68/0.84)	<0.0001

**Table 3 ijms-24-15665-t003:** Decongestion demonstrated by elastography in patients with Heart Failure.

Studies	Modality	Intervention	N	LSM before	LSM after	*p* Value
Millonig et al. [52]	FibroScan	Diuresis	10	40.7 (6.1–51.3) kPa	17.8 (3.3–33.2 kPa)	0.004
Colli et al. [53]	FibroScan	Diuresis	27	8.80 (5.92–11.90) kPa	7.20 (5.2–11.30) kPa	0.003
Hopper et al. [54]	FibroScan	Diuresis	8	11.2 (6.7–14.3) kPa	9.5 (7.3–21.6) kPa	>0.09
Alegre et al. [55]	FibroScan	Diuresis	9	14.7 (8.3–18.8) kPa	8.2 (5.1–11.2) kPa	0.008
Soloveva et al. [56]	FibroScan	Diuresis	149	12.2 (6.3–23.6) kPa	8.7 (5.9–14.4) kPa	<0.001
Yoshitani et al. [45]	ARFI	Diuresis	14	2.37 ± 1.09 m/s	1.27 ± 0.33 m/s	<0.001
Potthoff et al. [49]	ARFI	LVAD placement	23	1.88 (0.92–3.72) m/s	1.43 (0.93–3.67) m/s	<0.001
Sakamoto et al. [57]	Shear wave elastography	Diuresis	51	2.01 ± 0.61 m/s	1.62 ± 0.49 m/s	0.026
Pekoz et al. [58]	FibroScan	Atrial Septal Defect Closure	66	/	/	/

**Table 4 ijms-24-15665-t004:** Correlation between LSM and adverse outcomes.

Studies	Modality	Study Population	N	Cut-off for High LSM Group	Composite Endpoints	Number of Events	Mean FU Time (Days)	Independent Risk Factor (s) for Event
Saito et al. [60]	FibroScan	ADHF	105	>8.8 kPa (on admission)	Death from CVD; readmission for HF	42	153	LSM
Omote et al. [61]	pSWE/ARFI	ADHF	70	>1.50 m/s (on admission)	All-cause death; worsening HF	26	272	SBP LSM
Taniguchi et al. [62]	FibroScan	HF	171	>6.9 kPa (on discharge)	Cardiac death; readmission for HF	41	203	LSM
Soloveva et al. [56]	FibroScan	ADHF	149	>13 kPa (on admission) >5 kPa (on discharge)	All-cause death, heart transplant, or readmission for HF	71	289	LSM at discharge
Qian Wang et al. [63]	FibroScan	HF	53	>6.9 kPa (on discharge)	Death or readmission for HF	24	730	LSM TAPSE
Saito et al. [64]	US 2D-SWE	ADHFpEF	80	>10.2 kPa (on discharge)	All-cause death; readmission for HF	25	212	LSM
Zhang et al. [43]	FibroScan	Decompensated HFpEF	150	>8.30 kPa (on admission)	MACE (CV death, malignant arrhythmia, AMI, stroke, and rehospitalization for HF)	26	197	LSM AF NYHA class NT-proBNP
Panchani et al. [65]	US 2D-SWE	ADHF	49	>39.8 kPa (on admission)	LVAD, HT, death, and rehospitalization	21	365	LSM
de Ávila et al. [66]	FibroScan	Ambulatory HF (HFpEF, HFmrEF, HFrEF)	85	>5.9 kPa	CV death or HF hospitalization	20 HF hospitalizations 3 deaths	219 ± 86 days	LSM

## Data Availability

There are no new data associated with this article.

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
