# Peer review of "The Liver in Heart Failure: From Biomarkers to Clinical Risk"

_ijms, 2023, doi:10.3390/ijms242115665_

Round 1
Reviewer 1 Report
Comments and Suggestions for Authors
The paper from Aspromonte and CoAuthors focuses on persistence of right sided heart failure (HF) as a prominent marker of patient outcome putting on spot the liver congestion mirrored by liver stiffness (LSM) as a possible helpful index to be followed up in HF patients.
Currently precise right heart hemodynamic can be evaluated by performing inferior vena cava -echography or right heart catheterization during patient hospitalization, both procedures are not suitable after patient discharge and trustable jugular venous pressure detection can be a difficult assessment in daily practice.
On other hand the attempt to adopt specific liver markers in HF prognostication provided fuzzy information.
Recently the LSM has been proposed mostly by Japanese investigators as a reliable and non-invasive method to assess the persistency of right heart congestion in HF patients to lead therapy and prognosis as liver congestion relationship with current HF biomarkers has not been clearly defined.
In this setting LSM has been investigated to perform as a surrogate index of liver congestion mirroring right atrial pressure (RAP). By assuming this concept LSM should not be considered a direct RAP surrogate as it can express the intrinsic stiffness of hepatic parenchymal structure independent by the quantum of retained fluid.
By taking into account this concept LSM could be adopted as an index of RAP when baseline contemporary RAP (or central venous pressure) is detected at baseline.This concept should be expressed in the manuscript as it entails the need of contemporary measure of the two indexes to allow appropriate interpretation of baseline LSM value and of changes to decide timely therapeutic intervention. This point deserves to be addressed when the Author mention that “the therapeutic response with adequate decongestion during hospitalization may be an index of a better prognosis” (line 339).
As the liver fibrosis is the prominent unfavorable occurrence in subject suffering chronic liver disease and can be driven by a number of inflammatory/degenerative liver conditions as well as by chronic liver congestion, in the manuscript a specific diagnostic pathway should detail how to establish HF occurred without prior history of chronic liver disease. The point achieves paramount importance as the right sided HF is a marker of advanced disease prevailing in the older HF patient population, the one often bearing co-morbidities even liver related.
Author Response
Dear reviewer,
Thank you very much for your time.
We really appreciated the comments and tried to implement the manuscript, following the suggestions provided.
In detail:
- we implemented paragraph 3, to better clarify the usefulness of prognostic scores based on liver biomarkers in HF patients
- we implemented paragraph 5, underlining the utility of performing a measurement of LSM at least once in HF patients, simulteneously with the measurement of RAP in basal conditions. This would provide a reference value that could be used during any subsequent hospitalizations, comparing it with the new LSM measurement, in order to guide diuretic therapy.
- we implemented paragraph 2, underlining the importance of a correct clinical assessment of eventual comorbidities and alcohol consumption in patients with suspected CH.
We hope that with these changes the manuscript will now be more suitable for publication.
Thank you again
Reviewer 2 Report
Comments and Suggestions for Authors
This is an excellent review article on a timely topic, i.e., liver physiopathology during heart failure.
I have only a few comments to make, as detailed below.
1) Abstract and Introduction. I suggest the Authors spell out what the current review article is all about. This will guide the readers better.
2) Lines 49-55. Would it be possible for the Authors to generate a cartoon schematizing the main physiopathological changes in congestive hepatopathy?
3) Figures 1 and 2. Make sure the font size is big enough to be easily read.
4) Section 3 (Initial paragraphs). Insert some references in support of these contentions.
5) Section 4 (initial paragraphs). Ditto.
6) In invite the Authors to perform careful editing of the manuscript for typos and punctuation.
Comments on the Quality of English LanguageGood English, overall. However, punctuation needs attention.
Author Response
Dear reviewer,
Thank you very much for your time.
We really appreciated the comments and tried to implement the manuscript, following the suggestions provided.
In detail:
1) we implemented the abstract
2) we added a cartoon schematizing the main physiopathological changes in congestive hepatopathy
3) we modified the font size
4) we inserted some references
5) we inserted some references
We hope that with these changes the manuscript will now be more suitable for publication.
Thank you again